

# Development and validation of a nomogram to predict protein-energy wasting in patients with peritoneal dialysis: a multicenter cohort study

Ziwei Mei[1],[*], Bin Zhu[2],[*], Xiaoli Sun[1], Yajie Zhou[1], Yuanyuan Qiu[3], Xiaolan Ye[2], Hongjuan Zhang[2], Chunlan Lu[2], Jun Chen[4] and Hong Zhu[1]

[1] Lishui Municipal Central Hospital, Lishui, China
[2] Zhejiang Provincial People's Hospital, Hangzhou, China
[3] Longquan People's Hospital, Longquan, China
[4] Zhejiang Chinese Medical University, Hangzhou, China
[*] These authors contributed equally to this work.

Corresponding author
Hong Zhu, lszxyyzhuhong@163.com

## ABSTRACT

**Background:** Protein-energy wasting (PEW) is a common complication in patients with peritoneal dialysis (PD). Few investigations involved risk factors identification and predictive model construction related to PEW. We aimed to develop a nomogram to predict PEW risk in patients with peritoneal dialysis.

**Methods:** We collected data from end-stage renal disease (ESRD) patients who regularly underwent peritoneal dialysis between January 2011 and November 2022 at two hospitals retrospectively. The outcome of the nomogram was PEW. Multivariate logistic regression screened predictors and established a nomogram. We measured the predictive performance based on discrimination ability, calibration, and clinical utility. Evaluation indicators were receiver operating characteristic (ROC), calibrate curve, and decision curve analysis (DCA). The performance calculation of the internal validation cohort validated the nomogram.

**Results:** In this study, 369 enrolled patients were divided into development ($n = 210$) and validation ($n = 159$) cohorts according to the proportion of 6:4. The incidence of PEW was 49.86%. Predictors were age, dialysis duration, glucose, C-reactive protein (CRP), creatinine clearance rate (Ccr), serum creatinine (Scr), serum calcium, and triglyceride (TG). These variables showed a good discriminate performance in development and validation cohorts (ROC = 0.769, 95% CI [0.705–0.832], ROC = 0.669, 95% CI [0.585–0.753]). This nomogram was adequately calibrated. The predicted probability was consistent with the observed outcome.

**Conclusion:** This nomogram can predict the risk of PEW in patients with PD and provide valuable evidence for PEW prevention and decision-making.

## INTRODUCTION

PEW is a prevalent complication in chronic kidney disease patients with peritoneal dialysis, manifesting as a disorder of multiple metabolism and nutrition and a decrease of protein and fat (*Fouque et al., 2008*; *Orozco-González et al., 2022*). PEW results from inflammation, catabolic illness, and nutrient intake reduction. PD Patients with PEW develop a disequilibrium between energy intake and expenditure (*Kittiskulnam et al., 2021*; *Tennankore & Bargman, 2013*). According to the criteria of the International Society of Renal Nutrition and Metabolism (ISRNM), previous investigations reported PEW ranged between 14.1% and 83% in patients with peritoneal dialysis (*Liu et al., 2015*; *Markaki et al., 2016*; *Harvinder et al., 2016*; *Zhou et al., 2017*). PEW leads to lower quality of life and a higher hospitalization rate, morbidity, and mortality (*Velasquez et al., 2013*; *Yasui et al., 2016*; *Chao et al., 2017*). PEW is a crucial predictor of mortality in patients with peritoneal dialysis (*Wang, 2011*).

Peritoneal dialysis is an exchange between blood and a glucose-containing peritoneal solution. In this process, proteins and amino acids transport into the dialysate effluent. Patients lose amino acids of 1–3.5 g/d and proteins of 5–10 g/d during peritoneal dialysis, mainly albumin (*Kopple et al., 1982*; *Krediet et al., 1986*; *Westra et al., 2007*). A previous study presented that every 1 g/L elevation in serum albumin was independently related to an 8% reduction of all-cause mortality risk in patients with peritoneal dialysis. The subjective global assessment (SGA) score is a tool to estimate the nutritional condition. Every unit improvement in SGA score was associated with a 25% reduction of mortality risk in patients with peritoneal dialysis (*Wang, 2011*). Therefore, PEW prediction can prevent the loss of protein, energy, and mortality risk in patients with peritoneal dialysis. Nowadays, the ISRNM criterion is the only recognized method of PEW diagnosis. Several researchers supposed that this ISRNM criterion is complicated to diagnose PEW patients. Consequently, they aimed to develop more straightforward approaches to replace the ISRNM criterion for PEW diagnosis, such as SGA and Malnutrition Inflammation Score. However, these tools are not recognized methods for diagnosis applied in the clinical. In addition, they are merely used to estimate the nutritional status but hardly predict the risk of PEW (*Harvinder et al., 2016*). Predictive model development will guide clinicians to identify patients with a higher risk of PEW and prevent the occurrence.

Recently nomogram has been a technique for clinicians to predict the probability of disease or outcome. This study aimed to develop a PEW nomogram in patients with PD. According to the nomogram, clinicians can calculate PEW risk and identify the influence factors. Practical intervention approaches help to reduce PEW occurrence.

## METHODS

### Study design and participants

We retrospectively searched data on patients with PD from two hospitals in China. This study developed and validated a nomogram to predict PEW risk in patients with peritoneal dialysis.

We enrolled patients who had received maintenance peritoneal dialysis between January 2011 and November 2022. The enrolled criterion was the following set: (1) >18 years old; (2) with dialysis duration at least more than 3 months; (3) with follow-up visits and medical data records. We excluded patients unable to oral dietary intake or with multiple organ failure.

The Ethics Committee of Lishui Municipal Central Hospital has approved the study. The ethical approval number is 2021-111. We received written informed consent from participants in this study. Legally Authorized Representatives of illiterate participants provided informed consent for the study. The study was conducted following the Declaration of Helsinki.

## Procedure

We randomly divided the study cohort into training and validation sets with a proportion of 6:4. We collected twenty-six parameters from the peritoneal dialysis management system. These variables contained essential characteristics and clinical laboratory assessments. Basic features included gender, age, ≥65 years, dialysis duration, peritonitis occurrence number, Renin-Angiotensin-Aldosterone System inhibitors (RAASi) administration, and complications. Complications covered infection, peritonitis, hypertension, diabetes, and cardiovascular disease. Laboratory tests were ultrafiltration volume, KT/V, Ccr, BUN, Scr, serum potassium, sodium, calcium, phosphorus, CRP, glucose, TG, low-density lipoprotein (LDL), predialysis BUN, and PTH. PEW was the end-point. According to the PEW diagnosis proposed by ISRNM (*Harvinder et al., 2016*), we defined patients as PEW who accord with at least three of the four following criteria: (1) decreased serum albumin level of less than 3.5 g/L; (2) reduced body mass index (BMI) of less than 23 kg/m$^2$ or 10% total body fat; (3) wasted muscle defined by the lean tissue index, calculated as a lean tissue mass normalized to the height-squared in the 10th percentile of the reference population; and (4) low dietary protein intake.

## Parameters measure

This study examined some laboratory data from the last dialysis. PD patients infuse dialysate into the abdomen. After a dwell time of hours, patients discharge dialysis effluent to achieve a complete PD process. The ultrafiltration volume is the difference value of the dialysis effluent minusing the abdomen dialysate. PD patients have residual renal function and clearance performance from the dialysate. Therefore, Ccr in this research refers to a summation of the creatinine clearance rate from the kidney and dialysis. Before peritoneal dialysis, clinicians draw patients' blood to estimate serum creatinine levels. After dialysis, they collected urine and dialyzate to measure the creatinine values in these two samples. Clinicians conducted an enzymatic method to test the level of serum creatinine. The peritoneal dialysis management system installed a calculation formula. We obtained total Ccr results by importing creatinine values, urine and dialysate volume, and serum creatinine.

Researchers collected patients' blood to assess biochemical parameters, including plasma glucose, CRP, inflammatory indicators, tumor markers, *etc*. Imaging examination

and tumor markers evaluated if patients have cancer. Bone mineral density test estimated osteoporosis condition. The presence of at least two following points diagnoses patients as peritonitis (*As'habi et al., 2019*): (1) symptoms consistent with peritonitis, including abdominal pain or cloudy dialysis effluent; (2) dialysis effluent white cell count of more than 100/uL or $0.1 \times 10^9$/L (after a dwell time of at least 2 h), with polymorphonuclear leukocytes of more than 50%; (3) positive dialysis effluent culture. Normal blood pressure is a systolic pressure of 90–140 mmHg and a diastolic pressure of 60–90 mmHg. Patients over 140/90 mmHg under no medical administration are hypertensive. Patients consistent with at least one point can be diagnosed as diabetes: (1) fasting blood glucose more than 7.0 mmol/L; (2) 2-h postprandial blood glucose more than 11.1 mmol/L; (3) HbA1c ≥ 6.5%. We measured patients' height and weight to calculate BMI. The calculation formula is the weight divided by the height square. Magnetic resonance imaging, bio-electrical impedance analysis, and dual-energy X-ray absorptiometry test the total body fat and lean tissue mass. The lean tissue index resulted from lean tissue mass divided by the height square. Low dietary protein intake is a daily protein intake of less than 0.8 g/kg/d. Clinicians can trace patients' follow-up data in the peritoneal dialysis management system. This system recorded all essential information, biochemical parameters, and dialysis-related data in every examination.

## Statistical analysis

All data consisted of continuous and categorical variables. Continuous data were described by mean ± SD for normal distribution or median (IQR) for abnormal distribution. Categorical data were reported by percentage. T-test compared groups of normally distributed data. The Wilcoxon rank-sum test conducted a group comparison of abnormally distributed data. The Chi-square test analyzed the difference between categorical data. First, we screened predictors and developed a nomogram by multivariate logistic regression in the development set. They were presented as odds ratio (OR) with 95% confidence intervals (CIs). Second, we validated the nomogram in the training and test sets. ROC investigated the discriminated ability. The area under the curve (AUC) in development set for 0.75 or more indicated good discrimination. Calibration plots assessed the predictive accuracy. DCA estimated the clinical utility. All tests in this study were two-tailed tests. The $p \leq 0.05$ was confirmed statistically significant. The statistical analysis tool is STATA 15.0 (Stata Corporation, College Station, TX, USA).

## RESULTS

A total of 369 patients were enrolled in this study. They were divided into development ($n = 210$) and validation ($n = 159$) sets. A total of 184 (49.86%) of 369 patients were PEW. Baseline characteristics are shown in Table 1. The average age was 57.44 years. A total of 48.24% of patients were women. The median dialysis duration was 17.5 months in PEW patients and 24 months in non-PEW patients. There was no significant variable difference between the training and validation sets (Table 2). Datasets can be analyzed to screen predictors and construct a nomogram.

**Table 1 Baseline characteristics of participants with peritoneal dialysis.**

| Variable | PEW (*n* = 184) | Non-PEW (*n* = 185) | *p*-value |
|---|---|---|---|
| Gender | | | 0.714 |
| Male | 97 (52.7) | 94 (50.8) | |
| Female | 87 (47.3) | 91 (49.2) | |
| Age, year | 59.31 ± 11.74 | 55.58 ± 10.85 | 0.002 |
| Age > 65 | 66 (35.9) | 41 (22.2) | 0.004 |
| Dialysis duration (month) | 17.5 (8, 46.75) | 24 (11, 65) | 0.054 |
| Infection | 15 (8.2) | 20 (10.8) | 0.383 |
| Peritonitis | 10 (5.4) | 15 (8.1) | 0.307 |
| Peritonitis occurrence | 0 (0, 0) | 0 (0, 0) | 0.522 |
| Hypertension | 123 (66.8) | 113 (61.1) | 0.249 |
| Cardiovascular disease | 1 (0.5) | 1 (0.5) | 0.997 |
| Diabetes | 24 (13) | 26 (14.1) | 0.777 |
| BUN (mmol/L) | 20.42 ± 6.41 | 20.50 ± 10.17 | 0.929 |
| Predialysis BUN (mmol/L) | 29.9 (24.13, 36.88) | 26.8 (22.25, 35.35) | 0.047 |
| Serum creatinine (μmol/L) | 902.75 ± 264.71 | 985.82 ± 297.45 | 0.005 |
| Serum potassium (mmol/L) | 4.15 ± 0.69 | 4.01 ± 0.65 | 0.065 |
| Serum sodium (mmol/L) | 139.88 ± 2.92 | 140.17 ± 2.20 | 0.280 |
| Serum calcium (mmol/L) | 2.23 ± 0.21 | 2.30 ± 0.22 | 0.001 |
| Serum phosphorus (mmol/L) | 1.51 ± 0.40 | 1.58 ± 0.42 | 0.086 |
| CRP (mg/L) | 2 (1, 5) | 1 (1, 4) | 0.099 |
| Glucose (mmol/L) | 6.15 ± 2.21 | 6.55 ± 2.62 | 0.112 |
| TG (mmol/L) | 1.39 (0.98, 1.95) | 1.78 (1.17, 2.53) | <0.01 |
| LDL (mmol/L) | 2.66 ± 0.87 | 2.96 ± 3.11 | 0.211 |
| PTH (pg/mL) | 257.8 (130.18, 413.03) | 315.1 (155.2, 475.7) | 0.086 |
| RAASi | 89 (48.4) | 92 (49.7) | 0.79 |
| Ultrafiltration volume (L) | 300 (100, 650) | 450 (200, 800) | 0.002 |
| Ccr (umol/L) | 60.33 ± 16.14 | 62.55 ± 22.51 | 0.277 |
| Kt/V (ml/s/1.73 m$^2$) | 1.94 (1.74, 2.24) | 1.99 (1.78, 2.30) | 0.315 |

Notes:
BUN, blood urea nitrogen; CRP, C-reactive protein; TG, triglyceride; LDL, low-density lipoprotein; PTH, parathyroid hormone; RAASi, renin-angiotensin-aldosterone system inhibitors; Ccr, creatinine clearance rate; Kt/V, urea clearance index; PEW, protein energy wasting.
Categorical variables are showed as *n* (%). Continuous variables with normal distribution are reported as mean ± SD. Continuous variables with abnormal distribution are given as median (IQR).

## Predictors of PEW

Multiple logistic regression screened eight predictors associated with PEW (Table 3). These predictors were Scr (OR 0.9980, 95% CI [0.9965–0.9996], *p* = 0.012), age (OR 1.0346, 95% CI [1.0025–1.0677], *p* = 0.034), Ccr (OR 0.9589, 95% CI [0.9351–0.9834], *p* = 0.001), dialysis duration (OR 0.9871, 95% CI [0.9770–0.9974], *p* = 0.014), TG (OR 0.6764, 95% CI [0.4942–0.9259], *p* = 0.015), serum calcium (OR 0.2350, 95% CI [0.0609–0.9061], *p* = 0.035), glucose (OR 0.8163, 95% CI [0.6927–0.9618], *p* = 0.015), CRP (OR 1.0398, 95% CI [1.0001–1.0811], *p* = 0.050).

**Table 2 The characteristics of development and validation sets.**

| Variable | Development set ($n$ = 210) | Validation set ($n$ = 159) | $p$-value |
|---|---|---|---|
| Gender | | | 0.323 |
| Female | 106 (50.5) | 72 (45.3) | |
| Male | 104 (49.5) | 87 (54.7) | |
| PEW | 106 (50.5) | 78 (49.1) | 0.787 |
| Age, year | 57.34 ± 11.08 | 57.57 ± 11.92 | 0.853 |
| Age > 65 | 61 (29) | 46 (28.9) | 0.980 |
| Dialysis duration (month) | 22 (9, 51) | 18 (10, 48) | 0.451 |
| Infection | 23 (11) | 12 (7.5) | 0.269 |
| Peritonitis | 16 (7.6) | 9 (5.7) | 0.458 |
| Peritonitis occurrence | 0 (0, 0) | 0 (0, 0) | 0.880 |
| Hypertension | 139 (66.2) | 97 (61) | 0.304 |
| Cardiovascular disease | 2 (1.0) | 0 (0) | 0.217 |
| Diabetes | 30 (14.3) | 20 (12.6) | 0.635 |
| BUN (mmol/L) | 20.69 ± 9.72 | 20.15 ± 6.55 | 0.547 |
| Predialysis BUN (mmol/L) | 28.55 (22.38, 36.63) | 28.6 (23.3, 34.1) | 0.567 |
| Serum creatinine (μmol/L) | 955.62 ± 286.95 | 929.57 ± 280.92 | 0.384 |
| Serum potassium (mmol/L) | 4.10 ± 0.64 | 4.06 ± 0.72 | 0.595 |
| Serum sodium (mmol/L) | 140.08 ± 2.47 | 139.96 ± 2.74 | 0.652 |
| Serum calcium (mmol/L) | 2.27 ± 0.25 | 2.26 ± 0.16 | 0.493 |
| Serum phosphorus (mmol/L) | 1.56 ± 0.42 | 1.52 ± 0.40 | 0.414 |
| CRP (mg/L) | 2 (1, 4) | 2 (1, 5) | 0.828 |
| Glucose (mmol/L) | 6.43 ± 2.61 | 6.25 ± 2.17 | 0.490 |
| TG (mmol/L) | 1.53 (1.09, 2.33) | 1.54 (1.09, 2.21) | 0.667 |
| LDL (mmol/L) | 2.92 ± 2.93 | 2.67 ± 0.90 | 0.304 |
| PTH (pg/mL) | 271.8 (143.75, 445.78) | 278 (132.3, 461.1) | 0.859 |
| RAASi | 98 (46.7) | 83 (52.2) | 0.292 |
| Ultrafiltration volume (L) | 350 (150, 720) | 350 (110, 780) | 0.872 |
| Ccr (umol/L) | 61.29 ± 19.21 | 61.65 ± 20.16 | 0.860 |
| Kt/V (ml/s/1.73 m$^2$) | 1.99 (1.74, 2.30) | 1.94 (1.77, 2.23) | 0.453 |

**Notes:**
BUN, blood urea nitrogen; CRP, C-reactive protein; TG, triglyceride; LDL, low-density lipoprotein; PTH, parathyroid hormone; RAASi, renin-angiotensin-aldosterone system inhibitors; Ccr, creatinine clearance rate; Kt/V, urea clearance index; PEW, protein energy wasting.
Categorical variables are showed as $n$ (%). Continuous variables with normal distribution are reported as mean ± SD. Continuous variables with abnormal distribution are given as median (IQR).

## Nomogram of PEW in patients with peritoneal dialysis

We included eight predictors to develop a nomogram by multivariate logistic regression (Fig. 1). According to the model, we can identify the PEW risk for an individual patient. For example, a 65-year-old patient who experienced peritoneal dialysis for 40 months showed a recent laboratory examination: Ccr 60 ml/min, Scr 500 mmol/L, serum calcium 2.4 mmol/L, glucose 6 mmol/L, TG 3 mmol/L, CRP 5 mg/L. The corresponding score of each indicator was 26 points, 23 points, 70 points, 52 points, 36 points, 68 points, 50 points, and 3 points. His total score was 328 points. The risk of PEW was 50%.

Table 3 Multivariate logistic regression analysis in the development set.

| Variable | Multivariate logistic regression analysis | |
| --- | --- | --- |
| | OR (95% CI) | p |
| Scr | 0.9980 [0.9965–0.9996] | 0.012 |
| Age | 1.0346 [1.0025–1.0677] | 0.034 |
| Ccr | 0.9589 [0.9351–0.9834] | 0.001 |
| Dialysis duration | 0.9871 [0.9770–0.9974] | 0.014 |
| TG | 0.6764 [0.4942–0.9259] | 0.015 |
| Serum calcium | 0.2350 [0.0609–0.9061] | 0.035 |
| Glucose | 0.8163 [0.6927–0.9618] | 0.015 |
| CRP | 1.0398 [1.0001–1.0811] | 0.050 |

**Note:**
Ccr, creatinine clearance rate; Scr, serum creatinine; CRP, C-reaction protein; TG, triglyceride.

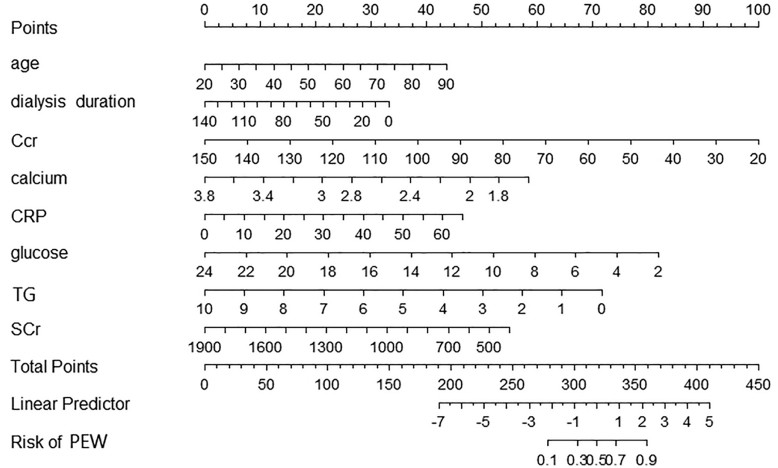

**Figure 1 A nomogram for PEW prediction in patients with peritoneal dialysis.** Each predictor level indicates a specific score. A summary of each predictor score generated a total score. The total score corresponds to PEW probability. PEW, protein-energy wasting.

## Nomogram evaluation

The Hosmer-Lemeshow test in development and validation sets was $\chi^2 = 10.42$, $p = 0.404$, and $\chi^2 = 13.06$, $p = 0.220$. This result presents the predicted outcomes were highly consistent with the actual effects. The ROCs in the development and validation sets performed a nomogram with an excellent discrimination ability (AUC 0.769, 95% CI [0.705–0.832], AUC 0.669, 95% CI [0.585–0.753]) (Figs. 2 and 3). We discovered a good correlation between predicted results and observed outcomes in the training and test sets (Figs. 4 and 5). The nomogram had a clinical utility from DCA (Figs. 6 and 7).

## DISCUSSION

In this study, the PEW incidence in patients with peritoneal dialysis was 49.86%. The rate was within the range reported in the previous study (*Harvinder et al., 2016*). For the first

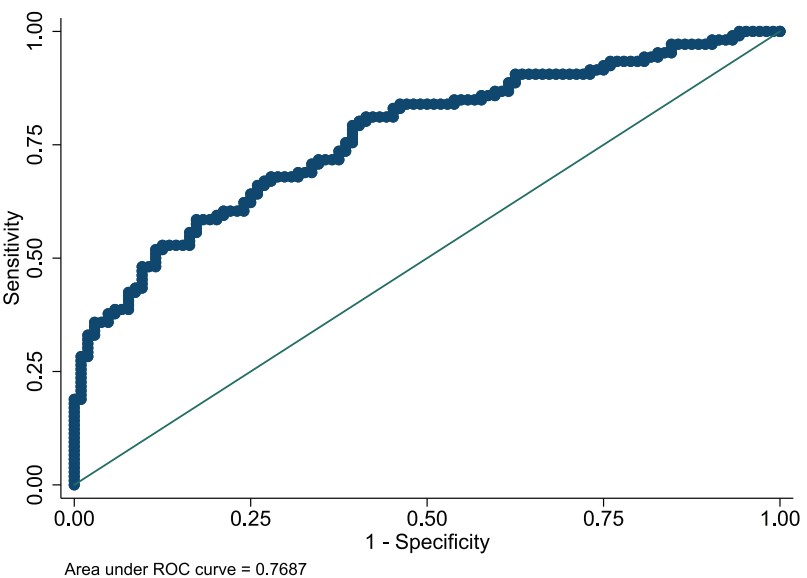

Area under ROC curve = 0.7687

**Figure 2 ROC curve of predictive model in the development set.** ROC, receiver operating characteristic; AUC, area under the curve.             

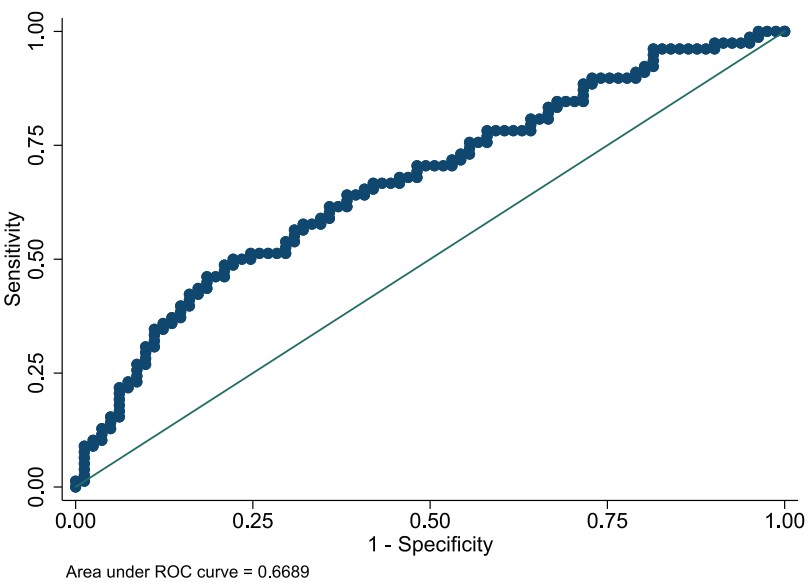

Area under ROC curve = 0.6689

**Figure 3 ROC curve of predictive model in the validation set.** ROC, receiver operating characteristic; AUC, area under the curve.             

time, this study developed a nomogram to predict PEW in patients with peritoneal dialysis. Predictors were age, dialysis duration, Ccr, Scr, serum calcium, CRP, TG, and glucose. This nomogram performed an excellent discrimination function and practice utility.
The predicted outcome highly correlated to the observed result.

Protein and amino acids transport to the dialysate in the peritoneal dialysis process. Patients with PD tend to suffer from protein catabolism, malnutrition, and inflammation. ISRNM suggested the status as PEW (*Harvinder et al., 2016*). The PEW incidence in

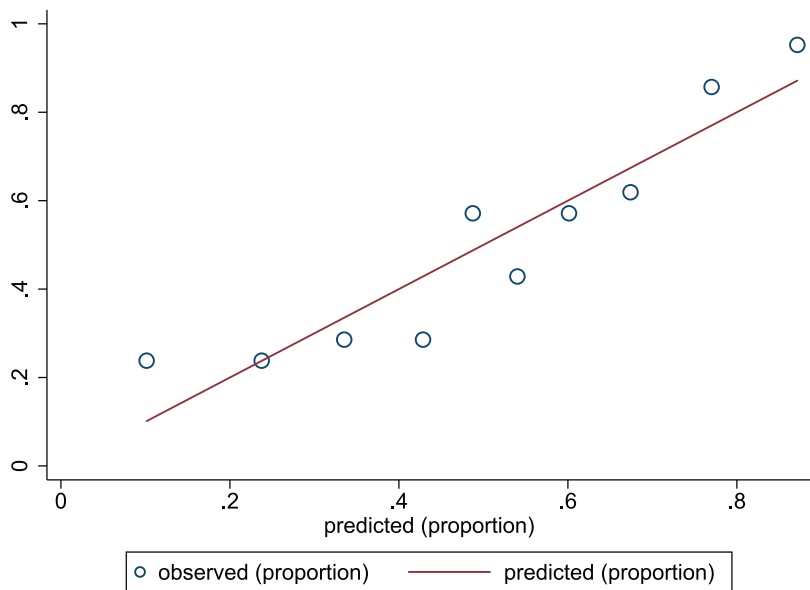

**Figure 4 Calibration plots of predictive model in the development set.** The x-axis is the predicted occurrence of PEW. The y-axis is the observed outcomes. The diagonal dotted line represents an ideal predictive outcome by an ideal model. The solid line represents the performance of nomogram. It better predicts that a solid line is close to a diagonal dotted line. The figure shows that nomogram have a good predictive ability.

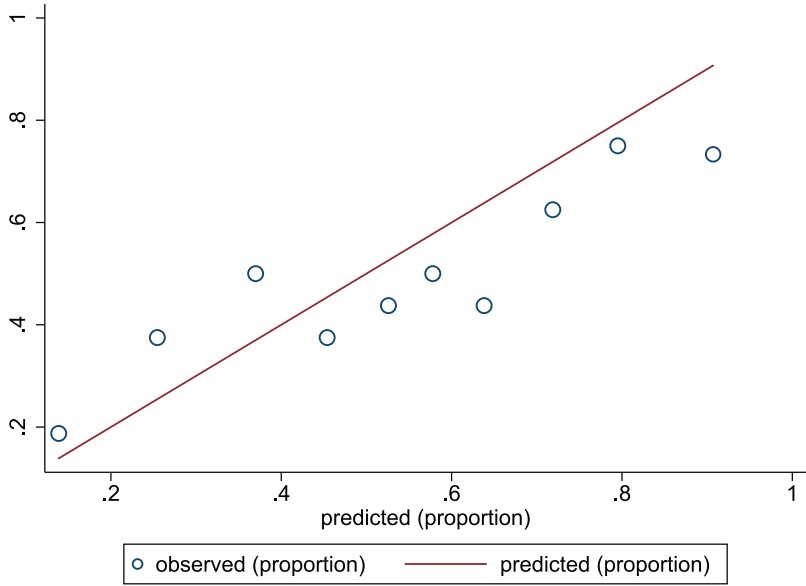

**Figure 5 Calibration plots of predictive model in the validation set.** The x-axis is the predicted occurrence of PEW. The y-axis is the observed outcomes. The diagonal dotted line represents an ideal predictive outcome by an ideal model. The solid line represents the performance of nomogram. It better predicts that a solid line is close to a diagonal dotted line. The figure shows that nomogram have a good predictive ability.

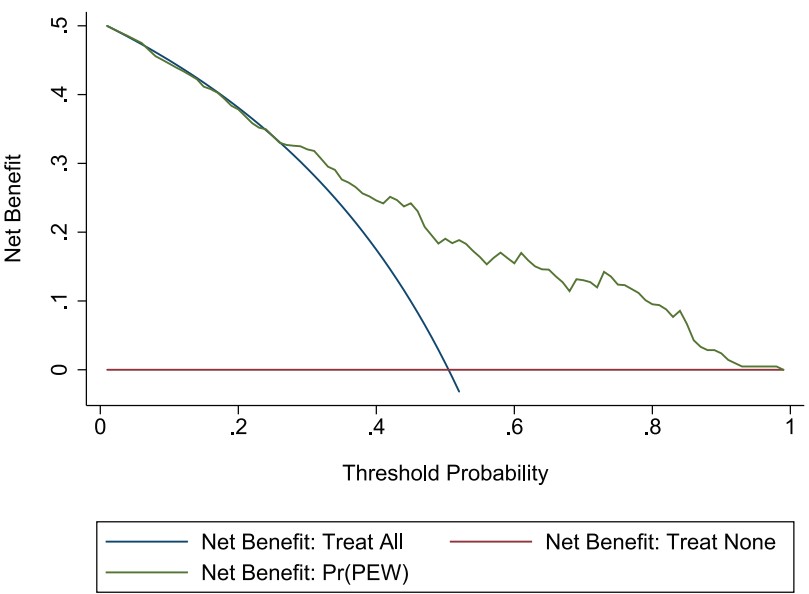

**Figure 6 DCA of the nomogram in the development set.** Red-solid line: the patient does not apply predictive model, and the net benefit is zero; blue-solid line: all patients are treated by nomogram. The area enclosed by the three lines demonstrates the clinical utility of nomogram. DCA, decision curve analysis.

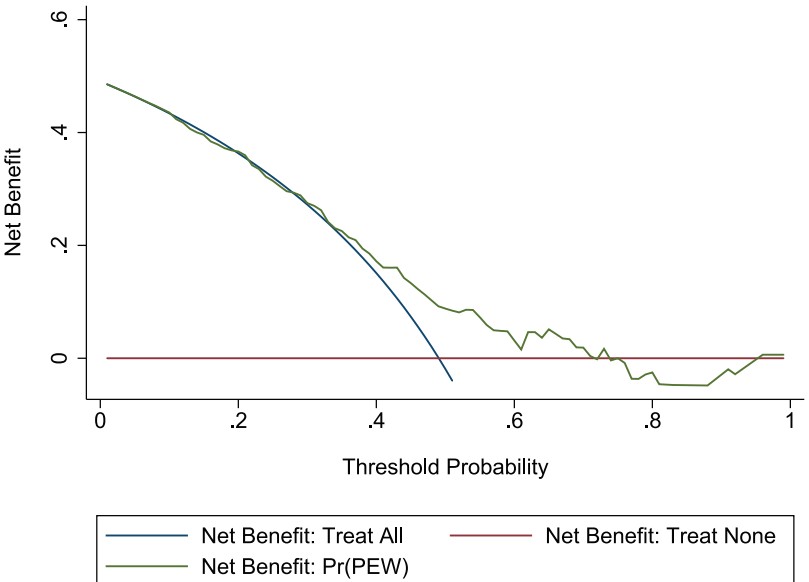

**Figure 7 DCA of the nomogram in the validation set.** Red-solid line: the patient does not apply predictive model, and the net benefit is zero; blue-solid line: all patients are treated by nomogram. The area enclosed by the three lines demonstrates the clinical utility of nomogram. DCA, decision curve analysis.

patients with PD is higher than in hemodialysis (*Harvinder et al., 2016*; *Kiebalo et al., 2020*). PEW increases rehospitalization rate, cardiovascular morbidity, and mortality (*As'habi et al., 2019*). Therefore, PEW prevention improves the prognosis of patients with PD (*Tsai et al., 2019*). This study recognized hypocalcemia as a risk factor for PEW.

Vitamin D modulates calcium absorption in the human body. Because vitamin D activation requires the 1,25(OH)2D in the proximal kidney tubule, renal injury weakens vitamin D activation to decrease serum calcium (*Cannata-Andía et al., 2021*; *Zappulo et al., 2022*). In CKD patients, hypocalcemia always results from vitamin D deficiency and indicates an occurrence of CKD-metabolic bone disease (CKD-MBD). Vitamin D and CKD-MBD are associated with PEW. Vitamin D has long been recognized as a contributor and involved in crucial molecular pathways for PEW (*Garcia et al., 2011*). Humans or experimental models with kidney injury presented a positive relationship between 25-hydroxyvitamin D and testosterone. This finding recommended that vitamin D may control muscle mass (*Lee et al., 2012*). Wasted muscle is one of the diagnostic criteria for PEW. Vitamin D insufficiency and CKD-MBD lead to bone loss and muscle decrease, which can predispose patients to PEW. Furthermore, hypocalcemia is a severe complication correlated to metabolic bone disease and bone loss, contributing to PEW (*Li et al., 2022*). Therefore, we should evaluate the potential vitamin D insufficiency and CKD-MBD in patients with hypocalcemia. Calcium supplements and CKD-MBD correction are effective methods to reduce PEW risk.

Although some variables' odds ratios were very close to 1, such as SCr, age, Ccr, dialysis duration, and CRP, we still considered them PEW predictors according to previous research and clinical observation. The crucial impact of inflammation on the nutrition condition has been published (*Carrero et al., 2007*; *Rattazzi et al., 2003*). Inflammation reduces energy and stimulates protein consumption in patients with peritoneal dialysis. CRP elevation is one evidence of inflammation, which can predict wasting in dialysis patients (*Landray et al., 2004*; *Kalantar-Zadeh et al., 2004*). A previous study discovered a negative correlation between CRP and protein, serum albumin, and serum pre-albumin (*Choi et al., 2010*). They reported CRP as a helpful marker. Regular CRP assessment can improve malnutrition management in patients with peritoneal dialysis. This nomogram indicated that every 10 mg/L elevation in CRP level would increase the PEW risk by 9.4% for patients with a total score of more than 275. In the clinical, the expected level of CRP is less than 10 mg/L. Patients with peritoneal dialysis should prevent infection and other situations leading to high CRP of more than 10 mg/L. Medical intervention reducing CRP plays a significant role in PEW prevention for patients with infection or inflammation.

Uremia promotes protein catabolism, increases the generation of non-nutritional toxins, and decreases albumin levels (*Garibotto et al., 2013*). Toxins clearance plays a vital role in patients with peritoneal dialysis. Ccr indicates the clearance ability. Most patients with peritoneal dialysis still have residual kidney function to clear toxins. Furthermore, peritoneal dialysis is a direct approach to help them eliminate toxins. Therefore, we summarized the Ccr of residual kidney function and peritoneal dialysis to estimate a patient's clearance ability. In this nomogram, every 10 ml/min reduction in total Ccr increased the PEW risk by 32% for patients with Ccr of less than 70 ml/min. Patients with Ccr of more than 70 ml/min prevent the PEW risk. However, ESRD patients will likely be confronted with several conditions reducing the residual kidney function and peritoneal dialysis performance. Maintaining a total Ccr level of more than 70 ml/min is challenging. In this study, the mean Ccr level is 61.44 ml/min. The number of patients with Ccr of more

than 70 ml/min was 101 of 369 (27.37%). Identifying and correcting factors reducing the Ccr level is essential to prevent PEW. Aged patients with peritoneal dialysis often experience loss of skeletal muscle mass. The changes in protein metabolism of patients with uremia were related to age (*Zhang & Ren, 2012*), a risk factor for PEW in patients with peritoneal dialysis. Clinicians should pay attention to PEW risk assessment for aged patients.

This study discovered that patients with short-term peritoneal dialysis had a higher PEW risk. No researcher reported the association between dialysis duration and PEW risk before. In the clinical, we found that patients with short-term peritoneal dialysis tend to present higher ultrafiltration. Higher ultrafiltration stimulated peritoneal protein loss, resulting in hypoalbuminemia (*Kiebalo et al., 2020*). Therefore, we considered that peritoneal protein losses probably elevate the PEW risk for patients with short-term peritoneal dialysis. A previous analysis indirectly supported this conclusion. They explored whether peritoneal protein transport increased with peritoneal dialysis treatment duration by collecting 103 patients treated with peritoneal dialysis for more than 4 years (*Goodlad & Davenport, 2017*). Researchers found higher dialysate/serum total protein ratios and reduced serum albumin in patients with initial exposure to peritoneal dialysis. This result indicated faster transport in patients who have just started peritoneal dialysis. With the prolongation of peritoneal dialysis duration, peritoneal creatinine, and total protein transport will change. Large pore transport for protein loss will not sustain an increase. Another report also revealed a consistent result. They proved the impact of peritoneal performance on patient characteristics at the start of dialysis. During long-term peritoneal dialysis, the peritoneal function can remain fundamentally steady (*Johansson & Haraldsson, 2006*). Therefore, patients with initial peritoneal dialysis likely occurred peritoneal protein loss because of better ultrafiltration and faster transport. Protein supplement was crucial for this kind of patient to prevent PEW.

Creatine (Cr) is an energy precursor essential to cellular physiology. In patients with chronic kidney disease, skeletal and cardiac muscles are depleted of Cr with the dialysis duration. Cellular damage due to Cr depletion lead to a deterioration of musculoskeletal and neurological function and poor quality of life. Cr supplements improve patients' musculoskeletal, brain, and peripheral nervous systems with chronic kidney disease. About 80–90% of total Cr exists in the human muscles. Approximately 2% of total Cr is converted to creatinine daily, which the kidney eliminates into the urine (*Wallimann, Riek & Möddel, 2017*). Serum creatinine is an indicator to measure the level of creatinine. PEW patients are always combined with muscle wasting, suggesting Cr reduction. Diminished conversion from Cr brought lower serum creatinine levels. Therefore, consistent with this nomogram, lower serum creatinine indicated a probability of muscle wasting, increasing the PEW risk.

## Limitation

There is some limitation in this study. First, participants in this nomogram are patients with peritoneal dialysis. This nomogram is not suitable for patients with hemodialysis. Second, we will continue to recruit more participants to achieve external nomogram validation.

## CONCLUSION

We developed a predictive model for PEW in patients with peritoneal dialysis. Predictors were age, dialysis duration, glucose, CRP, Ccr, Scr, serum calcium, and TG. This nomogram provided valuable evidence for predicting PEW in patients with peritoneal dialysis. According to this nomogram, clinicians can identify factors and make specific intervention methods to prevent PEW occurrence in high-risk patients.

## ACKNOWLEDGEMENTS

The author thanks the patients and doctors who participated in the research.

### Funding

This work is supported by funds from the Zhejiang Health Science and Technology Plan Project (2022RC091). The funders had no role in study design, data collection and analysis, decision to publish, or preparation of the manuscript.

### Grant Disclosures

The following grant information was disclosed by the authors:
Zhejiang Health Science and Technology Plan Project: 2022RC091.

### Competing Interests

The authors declare that they have no competing interests.

### Author Contributions

- Ziwei Mei conceived and designed the experiments, performed the experiments, analyzed the data, prepared figures and/or tables, authored or reviewed drafts of the article, and approved the final draft.
- Bin Zhu performed the experiments, prepared figures and/or tables, and approved the final draft.
- Xiaoli Sun conceived and designed the experiments, prepared figures and/or tables, and approved the final draft.
- Yajie Zhou conceived and designed the experiments, prepared figures and/or tables, and approved the final draft.
- Yuanyuan Qiu conceived and designed the experiments, authored or reviewed drafts of the article, and approved the final draft.
- Xiaolan Ye conceived and designed the experiments, authored or reviewed drafts of the article, and approved the final draft.
- Hongjuan Zhang performed the experiments, authored or reviewed drafts of the article, and approved the final draft.
- Chunlan Lu analyzed the data, authored or reviewed drafts of the article, and approved the final draft.

- Jun Chen performed the experiments, analyzed the data, prepared figures and/or tables, authored or reviewed drafts of the article, and approved the final draft.
- Hong Zhu conceived and designed the experiments, performed the experiments, prepared figures and/or tables, authored or reviewed drafts of the article, and approved the final draft.

## Human Ethics

The following information was supplied relating to ethical approvals (*i.e.*, approving body and any reference numbers):

The Ethics Committee of Lishui Municipal Central Hospital has approved the study. The ethical approval number is 2021-111.

## Data Availability

The raw measurements are available as a Supplemental File.

## Supplemental Information

Supplemental information for this article can be found online at http://dx.doi.org/10.7717/peerj.15507#supplemental-information.

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
