# Peer review of "Development and validation of a nomogram to predict protein-energy wasting in patients with peritoneal dialysis: a multicenter cohort study"

_PeerJ, doi:10.7717/peerj.15507_

## Round 0.1 · original submission · Major Revisions

The Methods section should be more detailed and the statistical analysis requires major changes before the manuscript can be taken into account.

·

Basic reporting

English language and style are fine, only minor spell check are required.
The introduction provide an appropriate background, nevertheless the influence of the different methods used for PEW diagnosis has to bettere explained (doi: 10.3389/fmed.2021.702749; doi: 10.1038/ejcn.2012.205).
In Table 4, the footnote is not well setted.
Figures are clear and useful to the reader.
The prevalence of PEW in this cohort study is high in front of other previous studies among peritoneal dialysis patients (doi: 10.1053/j.jrn.2020.11.007; doi: 10.3390/nu14163375), could the Authors hypothesize the cause of this?

Experimental design

The aim is an interesting one and relevant.
The methodological approach is appropriate for the Authors' aim and the research question is well defined and well described.

Validity of the findings

This paper would be an interesting update of the readers of Peer J and an useful tool for the clinician. The issue is not completely new, but it is still matter of debate.
All underlying data have been provided; they are robust, statistically sound, and controlled.
In the Discussion section, the message is clearly drive to the reader, and the conclusions correlate to the results found.

Additional comments

Thank you very much for the opportunity to review the manuscript. The aim is an interesting one and this paper would be an interesting update of the readers and an useful tool for the clinician.
The paper has to be improved by the Authors, only fixing some minor remarks.

Reviewer 2 ·

Basic reporting

Thank-you for the opportunity to review this manuscript which aims to identify predictors of PEW in patients on peritoneal dialysis. One of the strengths of the study is the relatively large n-number.

While the paper is readable in terms of language, there are many sentences which require review and don’t make sense. As an example (multiple others exist throughout the paper, which requires a review of language):
70 - 72: “However, these methods can’t predict the risk of PEW in patients with peritoneal dialysis. When these tools indicate the occurrence of abnormalities, patients always suffer from PEW.”
This sentence doesn’t make sense and is contradictory, it also requires referencing.

The figures and tables are incorrectly labelled with the numbering i.e. Table 1 etc. not matching between the script and the figures and tables.

Experimental design

More detail is required in the methods section.
How were the presence of peritonitis, hypertension, diabetes, cancer, AIDs and osteoporosis assessed?
How was ultrafiltration volume assessed?
How were laboratory tests conducted for example is this point of care glucose or laboratory plasma glucose. How was creatinine clearance assessed? How was serum creatinine measured (enzymatic or Jaffe methodology), were the measurements IDMS traceable?

What guidelines or recommendations are the diagnosis of PEW as stated based on?
How was total body fat assessed, how was lean tissue mass assessed?
How was low dietary protein intake defined?
When were parameters such as body mass index assessed, was this standardised for all patients?

Validity of the findings

Patients 370 – 416 (over 10% of the cohort) have identical values for Kt/V, creatinine clearance (despite major differences in serum creatinine), and ultrafiltration volume. This is not possible and draws serious concerns about the validity of the findings and statistical tests.

Many of the results don’t seem to correlate- there are patients with serum creatinine of over 1500 umol/L yet the creatinine clearance is 71 (this ties in with questions above- how was creatinine clearance determined). Similarly the coding for peritonitis and peritonitis times doesn’t make sense it seems that patients may have had peritonitis indicated under peritonitis times yet these patients are coded 0 for not having peritonitis (see also Table 1, which is also labelled Table 2 in the manuscript).

Some of the statements are not corroborated by evidence/ are untrue:
124 – 125: "There was no significant difference in variables between the training and validation sets (Table 2)." (As an example of mislabelling this table is labelled Table 3 on the document, Table 2 in the text and initial heading, and Table 1 above the table!)
Serum phosphate is significantly different. P = 0.024
146 – 148: "These predictors performed excellent ability of discrimination. The predicted outcomes by these biomarkers are highly correlated to observed results. Biochemical predictors selected by the study have practice utility." (Of note, the English needs correcting in these sentences)
The authors need to be honest about the true practical utility of their findings. Most of the odds ratios were very close to 1, variables such as: Scr, PTH and ultrafiltration volume only differed from 1 when the odds ratios were taken to 3 decimal places (a practice which wasn’t applied to other variables). Ultrafiltration volume odds ratio 95%CI includes 1, the authors should comment on why the p-value is still significant in this situation. The development and validation datasets are taken from one dataset which has been randomly split, this gives a false impression of the power of such findings. The authors need to acknowledge these short-comings and be consistent with their use of decimal places throughout the paper. A lot of the predictive power of the model is driven by the very large constant (or alpha) in the logistic regression model- which itself has a large standard error.
170 – 171: "This study indicated that patients with higher levels of CRP tend to develop an occurrence of PEW."
The study did not show this (the authors themselves acknowledge this in line 162).

When looking at the data the logit plot for Kt/V doesn’t appear to be linear, although this may be affected by the supposedly incorrect Kt/V values used (see first point in this section).

Additional comments

144: “The rate of PEW was higher than previous reports.” What is the reason for this, does this correspond to the rate in the patient populations or was it due to selection in the dataset. The authors should comment on if this affects the clinical utility of identified predictors.

A number of references are missing from the manuscript see as an example lines 152 – 159 where no references are given. I have indicated in parenthesis where references are missing: (further examples exist in the paper, especially in the discussion)
152 – 159: “The conception of PEW was suggested by ISRNM to describe decreased physical protein and fat due to inflammation, catabolic disease, and lower nutrient intake (REFERENCE). In chronic kidney disease patients with dialysis, the incidence of PEW is higher in peritoneal dialysis than in hemodialysis (REFERENCE). PEW increases rehospitalization rate, cardiovascular morbidity, and mortality (REFERENCE). Therefore, PEW prevention is important in the prognosis of patients with peritoneal dialysis. Previous researchers were barely involved in the risk factors of PEW. Few reports provide some valuable evidence and suggestion to identify patients with a high risk of PEW (REFERENCE).”

---

## Round 0.2 · accepted · Accept

Reviewers' suggestions were satisfactorily resolved by the authors. No further comments.

·

Basic reporting

I have no more comments to the Authors. The paper can be accepted to the present form.

Experimental design

The aim is an interesting one, and a rigorous investigation has been performed

Validity of the findings

Conclusions are well stated

Additional comments

The paper is well designed and written.